# Traditional Knowledge of the Utilization of Edible Insects in Nagaland, North-East India

**DOI:** 10.3390/foods9070852

**Published:** 2020-06-30

**Authors:** Lobeno Mozhui, L.N. Kakati, Patricia Kiewhuo, Sapu Changkija

**Affiliations:** 1Department of Zoology, Nagaland University, Lumami, Nagaland 798627, India; lnkakati@nagalanduniversity.ac.in (L.N.K.); patriciakiewhuo16707@gmail.com (P.K.); 2Department of Genetics and Plant Breeding, Nagaland University, Medziphema, Nagaland 797106, India; sapu@nagalanduniversity.ac.in

**Keywords:** *Antheraea assamensis*, *Apis cerana indica*, entomophagy, food, honey, Nagaland, preparation, *Samia cynthia ricini*, *Vespa mandarinia*, *Vespula orbata*

## Abstract

Located at the north-eastern part of India, Nagaland is a relatively unexplored area having had only few studies on the faunal diversity, especially concerning insects. Although the practice of entomophagy is widespread in the region, a detailed account regarding the utilization of edible insects is still lacking. The present study documents the existing knowledge of entomophagy in the region, emphasizing the currently most consumed insects in view of their marketing potential as possible future food items. Assessment was done with the help of semi-structured questionnaires, which mentioned a total of 106 insect species representing 32 families and 9 orders that were considered as health foods by the local ethnic groups. While most of the edible insects are consumed boiled, cooked, fried, roasted/toasted, some insects such as *Cossus* sp., larvae and pupae of ants, bees, wasps, and hornets as well as honey, bee comb, bee wax are consumed raw. Certain edible insects are either fully domesticated (e.g., *Antheraea assamensis*, *Apis cerana indica*, and *Samia cynthia ricini*) or semi-domesticated in their natural habitat (e.g., *Vespa mandarinia*, *Vespa soror*, *Vespa tropica tropica*, and *Vespula orbata*), and the potential of commercialization of these insects and some other species as a bio-resource in Nagaland exists.

## 1. Introduction

The world population is estimated to reach 9.8 billion by 2050 with a globally expected increase of 76% in the demand for animal protein [1]. Livestock production worldwide accounts for 70% of the agricultural land and, with the growing demand for meat and the declining availability of agricultural land, there is an urgent need to find additional or alternative sources of protein to provide for the increasing global population. Humans have harvested the eggs, larvae, pupae, and adults of certain insect species from the forest or other suitable habitats for thousands of years as part of regular diets, to stave off famine, to use therapeutically for medicinal and ritual purposes [2], and to try them as novelties [3]. The workshop in Chiang Mai (Thailand) in 2008 on “*Forest insects as food: humans bite back*” [4] emphasized edible insects as a natural food resource. To combat future food insecurity and to develop insects for food and feed, as first suggested by Meyer-Rochow [5], is now considered a viable strategy [6,7]. Insects may have originally been used as a snack or an emergency food item, but even today over 2100 known species of edible insects are still consumed by millions of people of 3071 ethnic groups in 130 countries [8,9,10,11]. Entomophagy is advocated as a source to combat future food insecurity mainly because of the insects’ abundance, high nutrient composition, high feed conversion efficiency, digestibility, and ease with which they can be bred [6,7,12,13,14]. Edible insects’ nutrient profiles are often very favourable with regard to dietary reference values and daily requirements for normal human growth and health. In marginalized societies, insect consumption has often bridged the gap between the availability and non-availability of conventional food items.

Nagaland is bestowed with natural resources and rich biodiversity supporting various plants and animals including a variety of insects associated with its natural vegetation. While conventional sources of animal-based protein are not always affordable to the rural people and also take a long time to become available, many insects are consumed as an alternative source of protein and as delicacies. The use of insects as food by the Nagas goes way back to those times when they were still called the “Naked people”. Insects such as beetles, dragonflies, grubs of white ant, grasshoppers, locusts, crickets, stink bugs, grubs of all sorts of bees and wasps, bee comb, and honey were used as food and played an important role in the diet of the different ethnic groups [15,16,17,18]. Most of the studies on entomophagy reported so far from Nagaland [19,20,21,22] are preliminary in nature and are restricted to a few insect species available in certain geographical areas of Nagaland. The present study is an extension of an earlier investigation in seven tribal communities of Nagaland by Mozhui et al. [23], providing a more detailed inventory and further documentation of the traditional knowledge of using edible insects as a promising and alternative food source for Nagaland.

## 2. Materials and Methods

### 2.1. Study Area

Nagaland is a state located in the north-eastern part of India covering an area of 16,579 km^2^. It is situated at 93°20′–95°15′ E and 25°6′–27°4′ N, in the confluence of East Asia, South Asia, and Southeast Asia. Considered one of the biodiversity hotspots (within the Indo-Burma region) of the world, the state has unique geographical location and varied altitudinal range. Out of the total geographical area, 85.43% (14,164 km^2^) constitutes the forest cover, of which 5137 km^2^ is dense and 9027 km^2^ is open forest [24]. Agriculture is the main economy of the state, which includes not only crop raising but all other allied activities such as animal rearing i.e., poultry, horticulture, pisciculture, sericulture, silviculture, livestock i.e., dairy cattle (buffalo, cow, mithun), goats, pigs, etc. Two types of farming systems—jhum or shifting cultivation and terrace or wet cultivation are practiced by the ethnic groups. Jhum cultivation is an extensive method of farming in which the farmers rotate land rather than crops to sustain livelihood [25]. Areas of jhum land are cleared once in five to eight years for better crop production. While in terrace cultivation, the entire hillside is cut into terraces and irrigated by a network of water channels that flows down from one terrace to the other and is easier to maintain than the jhum plots. However, due to the state’s wide altitudinal variation, terrace cultivation is found only in some rural pockets and majority of the population are engaged in shifting cultivation. Rice is the dominant crop and the main staple food of the Nagas.

### 2.2. Data Collection

The documentation is based on a four year field survey from 2014 to 2018 across 53 villages in Nagaland (Figure 1). The methodology followed in this study for data collection and insect identification is similar to that described in detail by Mozhui et al. [23]. Prior to survey, village heads were informed in advance for selection of informants for authentic documentation. Therefore, from each village, 6–8 informants comprising of village heads, traditional knowledge holders, edible insect farmers, edible insect collectors, educated youths, and homemakers were selected for the study. The survey was conducted only after getting ethical clearance from the village heads as well as the informants. Among the 370 informants interviewed, 248 were male and 122 were female, all belonging to the age group 25–104 years (Table 1). The documentation of edible insects in the present study is based on the responses collected from the informants with the help of a semi-structured questionnaire (See Appendix A). Informants were asked questions on the stages of edible insect consumption, mode of preparation of the insect species, seasonal availability, and preferences for selecting insect species for consumption. All the voucher specimens and photographs of edible insects are deposited at Department of Zoology, Nagaland University, Lumami, Nagaland.

## 3. Results and Discussion

### 3.1. Edible Insects in Nagaland

The present study contains an inventory of 106 insect species including 82 edible insect species recorded in our earlier work [23] that are regarded as health foods by the ethnic groups in Nagaland (Table 2). Belonging to 32 families and 9 orders, the percentage contribution of 106 edible insect species is: 24% Hymenoptera, 24% Orthoptera, 21% Hemiptera, 13% Coleoptera, 7% Odonata, 7% Lepidoptera, 2% Mantodea, 1% Isoptera, and 1% Diptera (Figure 2). Photographs of certain most frequently abundant and preferred species are depicted in Figure 3.

### 3.2. Consumption of Edible Insects

In Nagaland, as elsewhere [26,27] preferences in edible insect consumption are mainly due to six reasons: (a) availability of the insect species, (b) size of the insects, as generally larger insects are preferred for consumption, (c) taboos associated with the insect species, (d) one’s own palatability/taste preference, (e) market value of the insect species and, (f) traditional ethno-medicinal knowledge associated with the insect species. With the advancement in technology and availability of western foods, entomophagy practices are discontinued by the locals in many parts of the world [28] in the false belief that they would be more readily accepted as civilized and cultured individuals by representatives of the western world [5]. There are also risks that the cultural and ecological knowledge of entomophagy may get lost while adopting western dietary patterns [7]. However, in Nagaland, even though tribal people are exposed to modern food stuffs, edible insects are still acknowledged and continue to be an important food source providing nutrition and income especially to the rural poor.

People living in rural areas know very well which species to collect for consumption and this knowledge is acquired by their children, as in other countries with a tradition of consuming insects [29]. It is important to note that edible insects are not only an important source of protein but have ecological advantages over meat [30] and simultaneously aid in maintaining the diversity of habitats for other life forms by sustaining the local environment. The reported benefits of the human consumption of insects as an alternative to conventional food animals are numerous, including comparable levels of protein coupled with lower environmental impact due to lower emissions of greenhouse gases and lower land requirements during production [31,32]. Further, collection of edible insects (those considered as pests) for human consumption has a positive impact on the agricultural crops, being an alternative and efficient biological control method. Thus, consumption coupled with conservation of edible insects can benefit people as well as nature in an area and lead to a sustained use of this important bio-resource [33]. While the use value of different insect species has demonstrated that every Naga tribe has a preference of its own with regard to insect consumption [23], members of the Naga tribes are very specific with the way an insect is prepared for consumption such as boiling in a little amount of water (boiled), cooking with local spices/local ingredients (cooked), frying in hot oil (fried), cooking over fire or over hot charcoal (roasted), as chutney, or eaten raw. Local spices (garlic, ginger), fermented bamboo shoot, dried bamboo shoot, powdered fruits of *Rhus semialata* (Murray) are important local ingredients for preparation. Depending on one’s own palatability, the mentioned ingredients are added to enhance the flavour of the insect food, be it as main dish or replacement of conventional meat sources (e.g., beef, chicken, pork). Modes of consumption and preparation of different insects belonging to the nine orders are discussed below.

#### 3.2.1. Order Odonata

Although both adult and nymphal stages of dragonflies are consumed, the nymphs are greatly preferred (Figure 3a). Adult dragonflies are collected in large quantities during the months of July–October, while nymphs are collected mostly during the winter season (December–March). For consumption purpose, the dragon nymphs are prepared by boiling in a little amount of water and cooked until dry, while the adults (wings are removed) are fried and eaten as snacks.

#### 3.2.2. Order Orthoptera

Orthopterans are broadly grouped into (a) grasshopper and katydids, and (b) crickets.

##### Grasshoppers and Katydids

The Nagas are agriculturalist and look after extensive farmlands, where varieties of locusts and grasshoppers are commonly available. During harvest season (September–October), important edible insect species such as *Melanoplus bivittatus*, *Elimaea securigera*, *Hieroglyphus banian*, *Oxya fuscovittata* (Figure 4a), and *Oxya hyla* are collected in large quantities. The harvested surplus grasshoppers are sundried or smoked and preserved for future use. Grasshoppers are prepared as fried or are cooked with local spices (Figure 4b), while giant katydids (e.g., *Mecopoda nipponensis*, *Mecopoda elongata*, and *Pseudophyllus titan*) are roasted/toasted and eaten as snacks (Figure 4c). Some members of Naga tribe regard *Elimaea securigera* as a “health food” and prepare the insects as cooked or fried to be served as the main dish, replacing conventional meat sources (for e.g., beef, chicken, and pork).

##### Crickets

Crickets are commonly available from August to November and are generally handpicked or captured by various methods. For instance, field crickets (e.g., *Tarbinskiellus orientalis* and *Tarbinskiellus portentosus*) are dug out after pouring water inside their burrow making them float on the surface. The burrow of the cricket is easily recognisable, as they leave a mound of loose soil at the top of the burrow. Generally, a long stick is put in into the hole so as to keep track of the burrow; after which, the soil is removed slowly and the crickets residing in the burrow are collected. Other crickets such as *Acheta domesticus* and *Teleogryllus occipitalis* are light trapped and collected in large numbers at night. People also spread wheat bran along the paddy fields where crickets are most available. The crickets get attracted to the wheat bran and when they come to feed on it, they are either handpicked or trapped with nets. Crickets are generally preferred in cooked form (with bamboo shoots and local spices). However, some prefer them deep fried or boiled (Figure 5a–c). Besides cooking or frying, the traditional way of preparing crickets such as *Tarbinskiellus orientalis*, *Tarbinskiellus portentosus*, and *Teleogryllus occipitalis* are followed by some members of the Naga tribes. For example, the insects may be properly mixed with salt and dry bamboo shoot, stuffed inside cut bamboo pipes (approximately 30 cm in length) and closed with a banana leaf (the use of banana leaf is to enhance the flavour). The bamboo pipe is placed under the fire, and after 20–30 min the dish prepared is served as the main dish replacing the conventional meat sources.

#### 3.2.3. Order Mantodea and Isoptera

Although, the mantises (*Tenodera sinensis* and *Hierodula coarctata*) are consumed by only some members, the preferred way for consumption is toasting and they are eaten as snacks. Besides toasting, the head and the digestive tract are removed before preparation and then the insects are cooked with local spices until dry. Termite (*Macrotermes* sp. and *Odontotermes* sp.) are popular edible insect species of the Nagas with a high fat content [34] and are collected in plenty during the months of March–May and November–December. While some consumers prefer boiled termites, most prefer cooked or fried termites. When termites are collected in large quantities, they are sun dried to preserve them for longer use.

#### 3.2.4. Order Hemiptera

##### Stinkbugs

Depending on the species, stink bugs are available all throughout the year. For instance, the most preferred stink bugs such as *Coridius janus*, *Coridius singhalanus*, *Darthula hardwickii*, *Tessaratoma javanica*, and *Udonga montana* are found during the months of May–November. Stink bugs are mostly handpicked during the daytime. The red pumpkin stink bug *Coridius janus* (Fabr.) completes its developmental stages on its host plant (*Cucurbita moschata* Dutch) and is easily handpicked for consumption. They are also available on different bean plants such as string bean: *Phaseolus vulgaris* L.; cowpeas such as *Vigna sinensis* (Savi), *Vigna unguiculata* L.; and Goa bean: *Psophocarpus tetragonolobus* DC, etc.

Most of the stink bugs are preferred in their adult stage; however, the leafhopper (*Darthula hardwickii*) is preferred in its nymphal stage. Preference for consuming only the nymphal stage of *Darthula hardwickii* is mainly because of two reasons: (a) the nymphs are tastier than the adults and, (b) the strong pungent smell of the leafhopper once it gets matured is disliked. While the pentatomid bug *Eurostus grossipes* and the tessaratomid bug *Tessaratoma javanica* are preferred roasted, *Aspongopus nepalensis*, *Coridius janus*, *Coridius chinensis*, and *Coridius singhalanus* are preferred as chutney or cooked for consumption (Figure 6a). With *Udonga montana*, the freshly harvested insects are first boiled (boiling is done to reduce the strong pungent smell), and only then further prepared (with local spices and filtrate of fresh bamboo shoots) as chutney or fried for consumption (Figure 6b). During the month of April and May, *Udonga montana* species are collected in large quantities and people boil the insect and sun dry or smoke them for longer use.

##### Aquatic Hemiptera

Aquatic Hemipterans are commonly available during winter season from October to January. The giant water bug (*Lethocerus indicus*) and water scorpion (*Laccotrephes ruber*) are collected while fishing and are a delicacy among the Naga tribes. Adult *Lethocerus indicus* is popular for its enticing aroma and is consumed boiled, fried, or roasted (Figure 6c). Similarly, adult *Laccotrephes ruber* are cooked with local spices and fermented bamboo shoot and served as the main dish (Figure 6d).

##### Cicadas

Depending on the species, cicadas are found during the months from May to November. They are usually handpicked or collected with the help of bamboo pole (covered with sticky latex extract) and are trapped on the sticky ends of the bamboo pole [23]. Cicadas (*Tibicen pruinosa*, *Dundubia intemerata*, *Dundubia oopaga*, *Pomponia* sp., and *Pycna repanda*) are equally appreciated by the ethnic groups of Nagaland and are cooked or fried for consumption.

#### 3.2.5. Order Coleoptera

##### Wood and Timber Larvae

Most of the larval Coleoptera are available for consumption during the period from June to November. All wood or timber larvae are equally appreciated for their food value and are prepared by cooking (with local spices and fermented/dried bamboo shoot) and served as a main dish. While *Batocera rubus*, *Batocera parryi*, and *Batocera rufomaculata* are consumed only in their larval (Figure 7a,b) and pupal stage, *Anoplophora* sp. and *Orthosoma brunneum* are preferred at all stages (larva, pupa, and adult). Adult *Anoplophora* sp. and *Orthosoma brunneum* are roasted, while their larvae and pupae are cooked for consumption. Corm-borer (*Aplosonyx chalybaeus*) larvae and palm weevil (*Rhynchophorus ferrugineus*) larvae are cooked for consumption.

##### Aquatic Beetles

Aquatic beetles are available during the months from July to November and are collected during fishing. Predatory diving beetles (e.g., *Cybister limbatus*, *Cybister tripunctatus lateralis*, and *Sandracottus manipurensis*) and the phytophagous water beetle *Hydrophilus caschmirensis* are considered delicacies among all Naga tribes. For consumption, the elytra and membranous wings of the beetles are removed prior to cooking and then prepared by boiling, cooking, or frying (Figure 7c).

##### Scarab Beetles

Adult scarab beetles (*Xylotrupes gideon*, *Lepidiota stigma*, and *Holotricha* sp.) that are found during the months from June to September are consumed by a few tribes and are preferred as roasted/toasted for consumption. Scarab beetles are easily handpicked. For instance, *Lepidiota stigma* is usually found on litchi tree, *Litchi chinensis* Sonn.; the basic mode of collection is to shake the tree branches which makes the beetles fall down into the ground and they are then easily collected.

#### 3.2.6. Order Hymenoptera

Hymenopterans are differentiated as ants, bees, hornets, and wasps. Depending on the species Hymenopterans are available all throughout the year. Ants are mostly available for consumption from May to July; bees, especially *Apis cerana indica* are reared and available in every season, while hornets and wasps are most abundant for consumption from May to February.

Six (6) species of honeybees (Apis cerana indica, Apis dorsata dorsata, Apis dorsata laboriosa, Apis florea, Lepidotrigona arcifera, Lophotrigona canifrons) are commonly used as food in the region. Of all bee species, Apis cerana indica is the most important species as it is commonly available and can be reared or domesticated more easily than the other bee species. All bee stages (larva, pupa, adult) as well as the bees’ products are widely used for their food value as well as their medicinal value (Figure 8a,b). Honey is considered to be a strong and effective medical agent for treating cold, conjunctivitis, cough, diarrhoea, and pneumonia.

Besides bee and bee products, paper wasps (e.g., *Provespa barthelemyi*) and hornets (e.g., *Vespa soror*, *Vespa tropica tropica*, and *Vespa mandarinia*) are important edible insects (Figure 8c–h). Preferences in preparation of wasps (larva, pupa, and adult) depend on one’s own taste and the preparation is done by frying or cooking (Figure 8c). The giant hornet (*Vespa mandarinia*) is an important edible species mainly because the grubs are a delicacy, but it is difficult to capture and has a high market value (Figure 8d). For consumption, the larvae, pupae, and the adult are prepared by frying or cooking with local spices replacing the conventional meat sources (beef, chicken, pork etc.).

Different methods are followed to capture/collect the bees, wasps, and hornets. The common honeybee (*Apis cerana indica*) is collected by smoking. In the process, a piece of cloth is burnt, and the smoke is blown into the bees’ nest. The adults fly away and do not return until the smoke disappears. The rock honeybees *Apis dorsata dorsata* and *Apis dorsata laboriosa* are collected in two steps. Firstly, the beehive is pierced to allow honey to drip down; after which the honey is applied on the exposed body areas. This way, although the bees remain on the hive they do not sting the collector and are easily collected. The nest entrance of the underground stingless bee *Lophotrigona canifrons* is very small; therefore, a stick is inserted into the entrance so as to keep track of the nest. The bees’ nest and the bee products are then collected by digging the soil away.

Giant hornet of the species *Vespa mandariana* are collected by the meat trap method, but for other hornet species the smoking out method is used [23]. With only its honey balls considered edible, the violet carpenter bee (*Xylocopa violacea*) is preferred by only a few Nagas and is eaten raw. The weaver ant (*Oecophylla smaragdina*), which had its chemical composition analysed by Chakravorty et al. [34], is considered as an important food item by all members of the Naga tribes (Figure 8i). While, the immature stages (eggs, larvae, and pupa) of the ant are boiled or cooked (with local spices) for consumption, the adult ants are prepared as chutney in a mixture of chilly, salt, and dry fish ground to powdered form and are also preserved for longer use.

#### 3.2.7. Order Lepidoptera

Lepidopterans are mostly available for consumption from May to December. However, the eri silkworm (*Samia cynthia ricini*), which is reared widely in the region, is available for consumption throughout the year. The carpenter worm (*Cossus* sp.) with high protein and fat content, total phenol, and total antioxidant content [35] is the most popular and valued edible species, mainly because of its enticing aroma, its high market value, and ethno-medicinal properties. Although, some people prefer the larvae raw, most of the consumers prefer the larvae boiled or cooked with local spices served as a main dish on the menu (Figure 9a). The bamboo larvae (*Omphisa fuscidentalis*) are also an important edible insect that is preferred mainly for its crispness after being fried and eaten as snacks (Figure 9b). With *Omphisa fuscidentalis*, besides cooking or frying of the larvae, the traditional way of steaming is followed by some Nagas. For instance, the larvae (properly mixed with local spices) are wrapped with banana leaves that are then placed under warm ash. After 20–30 min, the packages are taken out and the larvae which are perfectly steamed are eaten as snacks.

The eri silkworm (larva and pupa) is relished by all Naga tribes and is prepared fried or cooked with local spices for consumption (Figure 9c,d). Tent caterpillars (*Malacosoma* sp.) are also a delicacy among the ethnic groups and are cooked or fried for consumption. The caterpillars are handpicked, when collected in plenty; the caterpillars are smoked for future use.

#### 3.2.8. Order Diptera

The only dipteran species consumed, i.e., *Tipula* sp. (larva) is an important edible species and is prepared by boiling or cooking (faeces and gut removed) with local spices and served as a main dish on the menu. The larvae are mostly available for consumption during winter season (September–January).

## 4. Rearing and Marketing of Edible Insects

Having a high feed conversion and low feed consumption ratio, coupled with fast production of protein as compared to conventional animal and plant sources [36], rearing of edible insects is found to be more advantageous in providing unconventional protein to humans. While most of the insects documented from Nagaland are collected from the wild, the eri silkworm, muga silkworm (*Antheraea assamensis*), and *Apis cerana indica* are reared for personal consumption as well as for marketing (Figure 10a,c). However, wasps (e.g., *Provespa barthelemyi*) and hornets (*Vespa affinis indosinensis*, *Vespa auraria*, *Vespa basalis*, *Vespa bicolor*, *Vespa soror*, *Vespa tropica tropica*, *Vespa mandarinia*, and *Vespula otrbata*) are semi-domesticated. All throughout the year, different kinds of edible insects and insect products are sold in the markets (Figure 10b,c). While most of the edible insect species that are sold in the markets are freshly harvested ones, different honeybee products are now packaged and sold at specialized stores. The Nagaland Beekeeping and Honey Mission (NBHM) set up by the Government of Nagaland implements programs and policies for the promotion and development of beekeeping in the state. NBHM-brand processed honey is commercially available; besides many local entrepreneurs are also putting in an effort to commercialize different bee products of Nagaland. Apart from honey, all the other edible insects documented in the present study are not packaged and cannot be kept for longer durations. Therefore, even though there is potentiality of edible insects to become a staple food, ways for packaging edible insects and increasing their shelf-life to promote them as food nationally and internationally is the need of the hour.

Traditional way of increasing their shelf-life is practiced by the local insect sellers by simply smoking or sun drying the insect species (Figure 10d–f). The traditional method is nothing but minimizing the moisture content of the insect to reduce the rate of deterioration. The pentatomid bug *Udonga montana* that is boiled and further smoked/sundried (maintains its crispness) is also available in the local markets. Boiling and steaming an insect (larva and adult) not only maintains its size and keeps the desired aroma and flavour but also gives a higher score in terms of crispness [37]. Food dehydration is one way for the commercialization of edible insects, while the other way is to turn them into pickles. All over the world, the main concern with edible insect consumption, collection, and marketing is overexploitation. Entomophagy is an age-old practice in Nagaland and the different ethnic groups believe in sustainable utilization of the edible insect species. For instance, while collecting different kinds of bees and wasps, the queen along with some workers is left behind in a brood to enable them for further development and production. This effort for sustainability has been followed through generations and till today apiculture is successful in the region.

## 5. Conclusions and Recommendations

For the different ethnic groups in Nagaland, edible insects are a natural, renewable resource that plays an important role in providing nutrition. Currently, edible insects are mainly harvested from the wild and sometimes from agricultural crops where they occur as pests. However overharvesting edible insects from the wild to supply to the urban market for improving livelihood is not sustainable and is a strain to the rural biodiversity. Therefore, while edible insects have the potential to improve rural livelihood, semi-cultivation and farming of insects has to be seen as a priority. Looking into the Indian scenario, Nagaland has the potentiality of commercializing insects as a bio-resource. At present, only bees (especially *Apis cerana indica*), eri silkworm, and muga silkworm are mass produced in Nagaland. However, besides the mentioned three edible insects, there are groups of insects e.g., crickets, giant water bug, water beetles, palm weevil larvae, grasshoppers, etc. that are successfully reared in other parts of the world [38,39]. Nagaland harbours large number of edible insects such as aquatic bugs and beetles (*Cybister limbatus*, *Cybister tripunctatus lateralis*, *Lethoceruus indicus*, *Hydrophilus cashmirensis*) bees (*Apis cerana indica*, *Apis dorsata dorsata*, *Apis dorsata laboriosa*, etc.), crickets (*Tarbinskiellus orientalis*, *Tarbinskiellus portentosus*, *Teleogryllus occipitalis*), grasshoppers (*Oxya fuscovittata* and *Oxya hyla*), palm weevil (*Rhynchophorus ferrugineus*) and wasps (*Vespa mandarinia*, *Vespa soror*, *Vespa tropica tropica*, *Vespula orbata*, etc.), which have the potential for mass production in the region.

A study undertaken by the International Livestock Research Institute (ILRI), reported an increasing demand for livestock products and the present state production is not enough to feed the growing population. As per sample survey report of 2017–2018, the state produced 45.23% of total requirement of livestock worth 1206.15 crore Indian rupees but leaving a shortfall of 54.77%. Out of this shortfall, the state imported animal husbandry products worth 212.05 crore Indian rupees in monetary terms. In the backdrop of the growing demand for meat and the declining availability of agricultural land in the state, edible insects could be the alternative source of protein. Therefore, the present study, not only raises awareness among the tribal communities but also develops deep interest amongst policy makers and stake holders of the potential growth of the edible insect sector and thereby to promote funding into edible insect research and development. We believe that mass production, proper commercialization, and marketing strategies can improve livelihoods of tribal communities (especially the womenfolk) living in remote villages.

## Figures and Tables

**Figure 1 foods-09-00852-f001:**
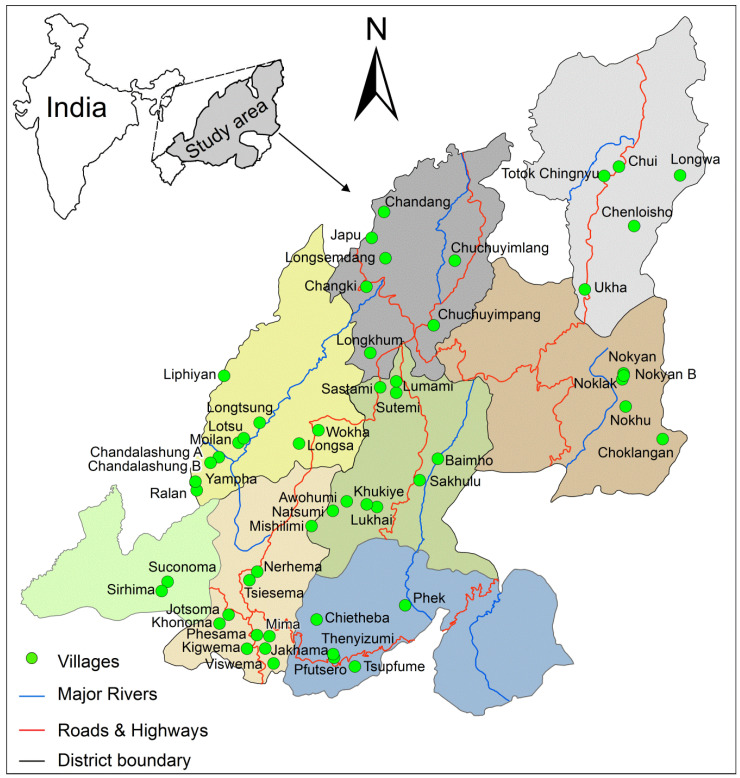
Location map of 53 villages surveyed for the study (Source: Survey of India toposheets).

**Figure 2 foods-09-00852-f002:**
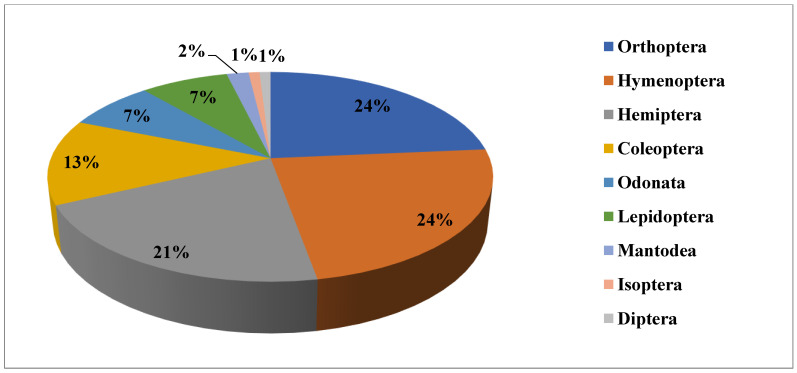
Percentage contribution of edible insects by each insect order.

**Figure 3 foods-09-00852-f003:**
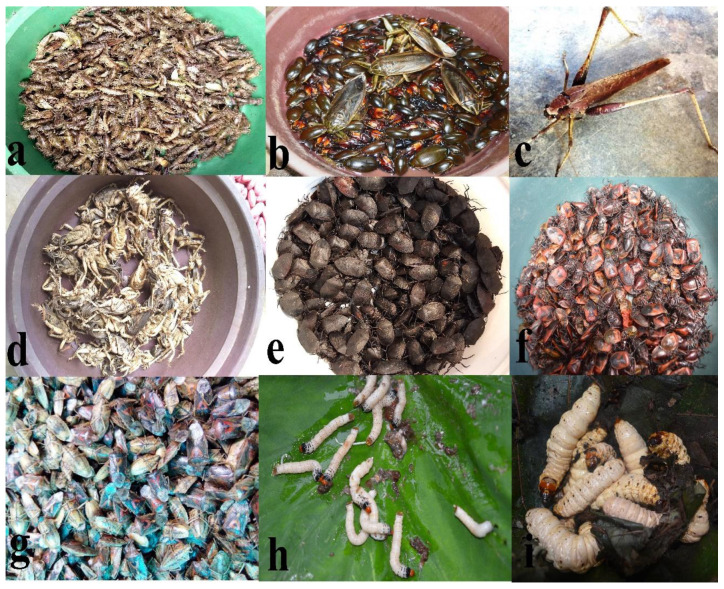
(**Plate 1**). Certain popular edible insect species of Nagaland. (**a**) Dragon nymphs, (**b**) various aquatic insects, (**c**) katydid *Mecopoda elongata*, (**d**) the sand cricket (*Schizodactylus monstrosus*), (**e**) the dinorid bug *Coridius singhalanus*, (**f**) the red pumpkin bug *Coridius janus*, (**g**) sundried pentatomid bug *Udonga montana*, (**h**) the chrysomelid beetle *Aplosonyx chalybaeus* larvae, and, (**i**) large unidentified wood larvae.

**Figure 4 foods-09-00852-f004:**
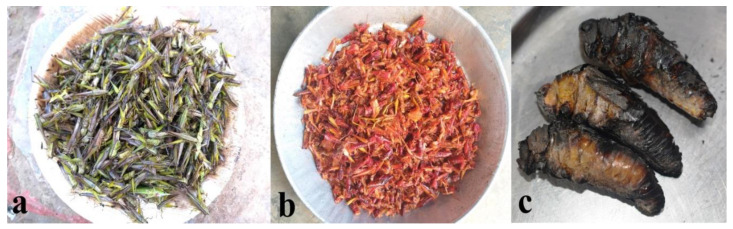
(**Plate 2**). Commonly utilized Orthopterans; (**a**) freshly harvested grasshoppers (*Oxya fuscovittata*), (**b**) grasshoppers cooked with fermented bamboo shoot, (**c**) roasted giant katydid (*Pseudophyllus titan*).

**Figure 5 foods-09-00852-f005:**
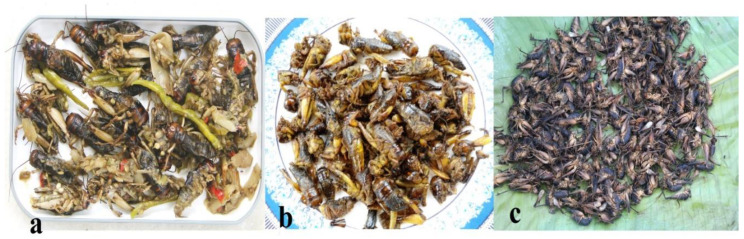
(**Plate 3**). Preferences in preparation of important cricket species; (**a**) crickets cooked in fresh bamboo shoot, a delightful delicacy, (**b**) fried crickets, a much sought after delicacy, (**c**) boiled cave crickets for consumption.

**Figure 6 foods-09-00852-f006:**
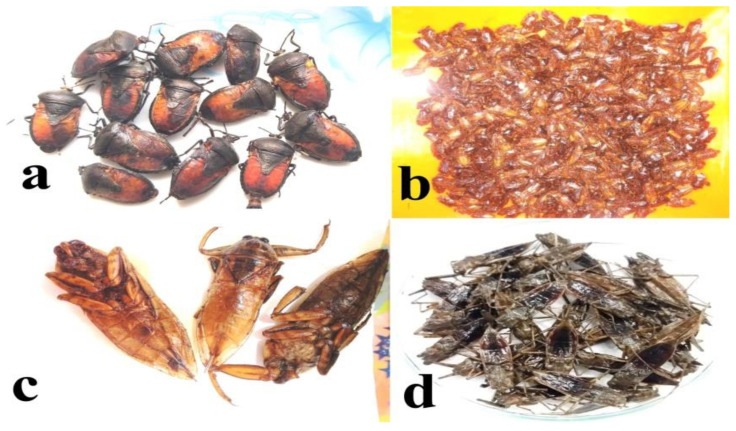
(**Plate 4**). Mode of preparation of various hemipterans: (**a**) roasted stink bugs (*Coridius singhalanus*), (**b**) fried pentatomid bug (*Udonga montana*), (**c**) fried giant water bugs (*Lethocerus indicus*), (**d**) boiled water scorpions (*Laccotrephes ruber*).

**Figure 7 foods-09-00852-f007:**
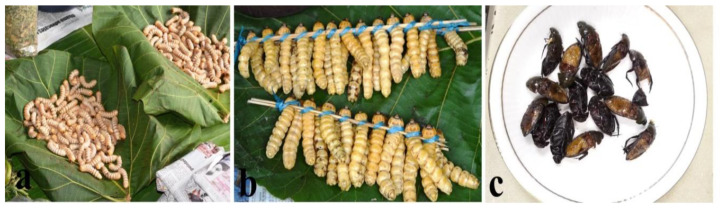
(**Plate 5**). Commonly available and consumed Coleopterans; (**a**,**b**) freshly harvested wood larvae sold at local market, Kohima, Nagaland, (**c**) fried water beetles (*Hydrophilus caschmirensis*).

**Figure 8 foods-09-00852-f008:**
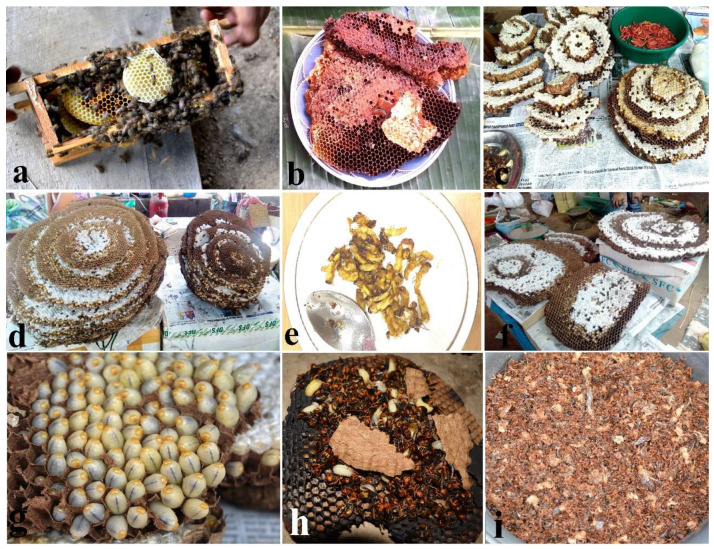
(**Plate 6**). Widely consumed and marketed Hymenopterans; (**a**) *Apis cerana indica* honey comb, (**b**) honeycomb of *Apis florea*, (**c**) wasp nest of *Vespa soror*, (**d**) large nests of *Vespa tropica tropica*, (**e**) fried wasps (*Provespa barthelemyi*), (**f**) nest of giant hornet (*Vespa mandarinia*), (**g**) healthy hornet grubs, (**h**) freshly harvested *Vespa auraria* by smoking method, and (**i**) worker ants *Oecophylla smaragdina*.

**Figure 9 foods-09-00852-f009:**
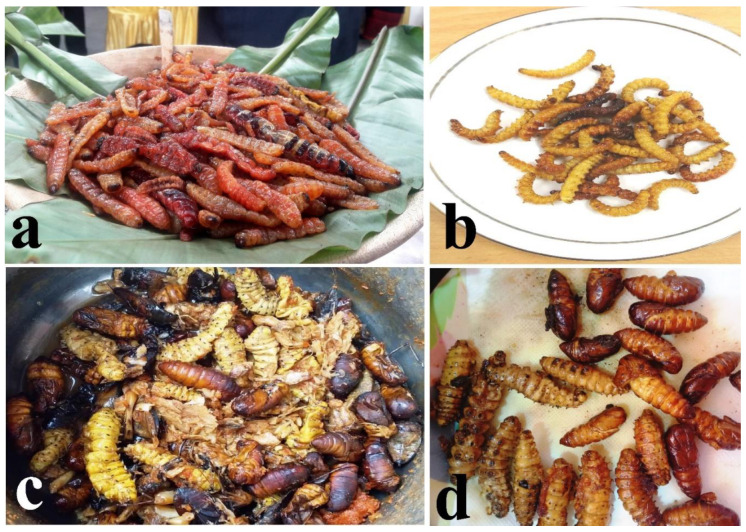
(**Plate 7**). Mode of preparation of important Lepidopterans; (**a**) fried carpenter worms (*Cossus* sp.), (**b**) fried bamboo caterpillars (*Omphisa fuscidentalis*), (**c**) silkworm larvae and pupae cooked with fermented bamboo shoot and local spices, (**d**) fried silkworms.

**Figure 10 foods-09-00852-f010:**
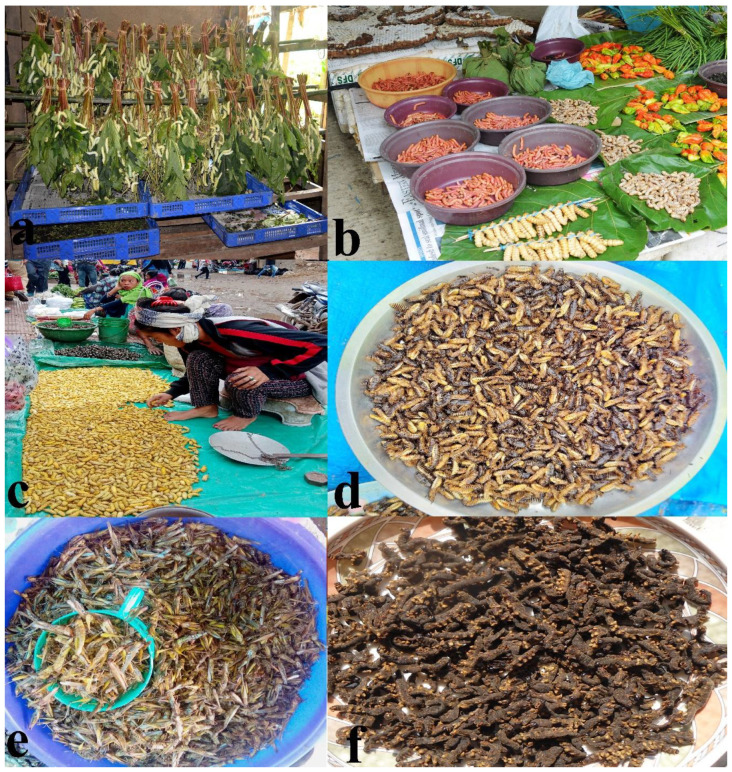
(**Plate 8**). Local insect rearing and various marketed edible insects; (**a**) rearing of eri silkworm (*Samia cynthia ricini*) at Socūnoma village, Dimapur district, (**b**) different species of wood larvae sold at local market, Kohima district, (**c**) a woman selling eri silkworms at local market, Dimapur district, (**d**) sundried termites sold at local market, Mokokchung district, (**e**) sundried grasshoppers sold at local market, Dimapur district, (**f**) smoked tent caterpillars sold at local market, Mokokchung district.

**Table 1 foods-09-00852-t001:** Demographic patterns of informants in the study area.

Gender	Number of Informants
Male	248 (67%)
Female	122 (33%)
**Age Group**	
25–34	60 (16%)
25–44	58 (16%)
45–54	59 (16%)
55–64	55 (15%)
65–74	57 (15%)
75–84	58 (16%)
85–94	18 (5%)
95–104	5 (1%)
**Educational status**	
Below high school	230 (62%)
Above high school	140 (38%)
**Informant status**	
Key informant	198 (54%)
General informant	172 (46%)

**Table 2 foods-09-00852-t002:** Insect species utilized as food by different ethnic groups in Nagaland (adapted from Mozhui et al., 2017) with new addition of 24 species (*).

Order	Family	Scientific Name	Local Name	Seasonal Availability	Edible Stage	Mode of Consumption
Odonata	Libellulidae	*Diplacodes trivialis* Rambur	Tukhakupu	July–Nov	nymph	B, C
		*Orthetrum pruinosum neglectum* Rambur	Tukhakupu	July–Nov	nymph	B, C
		*Neurothemis fulvia* Drury	Tukhakupu	July–Nov	nymph	B, C
		*Crocothemis servilia servilia* Drury	Tukhakupu	July–Nov	nymph	B, C
		*Potamarcha congener* Rambur	Tukhakupu	July–Nov	nymph	B, C
		*Orthetrum sabina sabina* Drury	Tukhakupu	July–Nov	nymph	B, C
		*Orthetrum triangulare triangulare* Sely	Tukhakupu	July–Nov	nymph	B, C
		*Pantala flavescens* Fabr.	Tukhakupu	July–Nov	nymph	B, C
Orthoptera	Acrididae	*Acrida exaltata* Walker	Tsütheku	June–Oct	adult	C
		*Atractomorpha* sp.	Anishe	June–Oct	adult	C
		**Choroedocus robustus* De Geer	Shenaqhu	June–Oct	adult	C
		*Chondracris rosea* Serville	Kuprie	June–Oct	adult	RO/TO
		**Gastrimargus africanus africanus* Saussure	Tluqhu	June–Oct	adult	C
		*Hieroglyphus banian* Fabr.	Sapathika	June–Oct	adult	C
		*Melanoplus* sp.	Yeghutukha	June–Oct	adult	C
		*Melanoplus bivittatus* Say	Shenaqhu	June–Oct	adult	RO
		*Oxya hyla* Serville	Naghilithika	June–Oct	adult	C
		**Oxya fuscovittata* Marschall	Naghilithika	June–Oct	adult	C
		**Phlaeoba infumata* Brunner	Atikha	June–Oct	adult	C
	Gryllidae	*Acheta domesticus* Linn.	Achuqu	Aug–Nov	adult	C
		*Gryllus* sp.	Ashuko	Sept–Nov	adult	C
		*Meloimorpha cincticornis* Walker	Petu	Aug–Nov	adult	C
		**Teleogryllus* sp.	Amishibachuqu	Aug–Nov	adult	C
		**Teleogryllus occipitalis* Serville	Awusho	Aug–Nov	adult	C
		*Tarbinskiellus orientalis* Fabr.	Awusho	Aug–Nov	adult	C
		**Tarbinskiellus portentosus* Lichtenstein	Awusho	Aug–Nov	adult	C
	Gryllotalpidae	*Gryllotalpa orientalis* Burmeister	Alhaqu	Aug–Nov	adult	C
	Tettigoniidae	*Mecopoda nipponensis* Haan	Kaghalapu	June–Oct	adult	RO/TO
		*Mecopoda elongata* Linn.	Khotsule	June–Oct	adult	RO/TO
		*Pseudophyllus titan* White	Salhatiti	June–Oct	adult	RO/TO
		**Elimaea securigera* Brunner	Anishe	June–Oct	adult	C
Orthoptera	Tettigoniidae	*Tettigonia* sp.	Anishe	June–Oct	adult	C
	Schizodactylidae	**Schizodactylus monstrosus* Drury	Awusho	June–-Aug	adult	C
Mantodea	Manitidae	*Tenodera sinensis* Saussure	Tsukole	July–Sept	adult	RO
		*Hierodula coarctata* Saussure	Kupkamichukole	July–Sept	adult	RO
Isoptera	Termitidae	*Macrotermes* sp.	Alho	Nov–Dec	adult	C
Hemiptera	Aetalionidae	**Darthula hardwickii* Gray	Thezü	Sept–Oct	nymph	C, F
	Belostomatidae	*Lethocerus indicus* Lepeltier and Serville	Tsungosho	Nov–Jan	adult	C
	Cicadidae	*Tibicen pruinosa* Say	Akoko	May–Nov	adult	C
		*Dundubia intemerata* Walker	Akoko	May–Nov	adult	C
		*Dundubia oopaga* Distant	Akoko	May–Nov	adult	C
		*Pomponia* sp.	Cievü	May–Nov	adult	C
		*Pycna repanda* Amyot and Serville	Cievü	May–Nov	adult	C
	Coreidae	*Dalader planiventris* Westwood	Akhane	May–July	adult	C
		*Anoplocnemis phasiana* Fabr.	Akhane	May–Aug	adult	C
		*Notobitus meleagris* Fabr.	Akhane	Aug–Sept	adult	C
	Dinidoridae	*Aspongopus nepalensis* Westwood	Akhane	Dec–Jan	adult	C
		*Coridius janus* Fabr.	Akhane	May–July	adult	C, CH
		*Coridius chinensis* Dallas	Akhane	May–July	adult	C, CH
		**Coridius singhalanus* Dist.	Akhane	Feb–Mar	adult	C, CH
	Nepidae	*Laccotrephes ruber* Linn.	Akhane	Oct–Jan	adult	C
	Pentatomidae	*Chinavia hilaris* Say	Akhane	May–July	adult	C
		*Cyclopelta siccifolia* Westwood	Akhane	May–July	adult	C
		*Dolycoris* sp.	Akhane	April–May/Nov	adult	C, CH, F
		*Erthesina fullo* Thunberg	Akhane	Aug–Sept	adult	C
		*Eurostus grossipes* Dallas	Akhane	May–July	adult	RO
Hemiptera	Pentatomidae	**Udonga montana* Distant	Akhane	April–May/Nov	adult	C, CH, F
	Tessaratomidae	*Tessaratoma javanica* Thunberg	Akhane	May–July	adult	RO
Coleoptera	Cerambycidae	*Anoplophora* sp.	Nukuo	Sept–Jan	all	C, RO
		*Batocera rubus* Linn.	Akulho	June–Nov	larva, pupa	C
		*Batocera parryi* Hope	Akulho	June–Nov	larva, pupa	C
		**Batocera rufomaculata* De Geer	Akulho	June–Nov	larva, pupa	C
		*Orthosoma brunneum* Forster	Akulho	July–Nov	all	C, RO
	Chrysomelidae	**Aplosonyx chalybaeus* Hope	Akallakulho	Sept–Nov	larva	C
	Curcurlionidae	*Rhynchophorus ferrugineus* Olivier	Akulho	Oct–Nov	all	C
	Dytiscidae	*Cybister limbatus* Fabr.	Azükhanu	July–Nov	adult	C
		**Cybister tripunctatus lateralis* Fabr.	Azükhanu	July–Nov	adult	C
		**Sandracottus manipurensis* Vazirani	Dzübolo	July–Nov	adult	C
	Hydropillidae	*Hydrophilus caschmirensis* Redtenbacher	Azühpüllau	July–Nov	adult	C
	Scarabaeidae	*Xylotrupes gideon* Linn.	Shushu	June–Aug	adult	RO
Coleoptera	Scarabaeidae	*Lepidiota stigma* Fabr.	Kolompvu	Aug–Sept	adult	RO
		**Holotricha* sp.	Befu	July–Aug	adult	RO
Hymenoptera	Apidae	*Apis cerana indica* Fabr.	Aghui	Whole year	all	R, C
		*Apis dorsata dorsata* Fabr.	Ati-i	Sept–May	all	R, C
		*Apis dorsata laboriosa* Smith	Atukhi	Sept–May	all	R, C
		*Apis florea* Fabr.	Aku-u	Sept–Feb	all	R, C
		*Lepidotrigona arcifera* Cockerell	Amgho	Sept–Feb	all	R
		*Lophotrigona canifrons* Smith	Tyita	Sept–Feb	all	R
		*Xylocopa violacea* Linn.	Khvukhvu	May–Oct	honey	R
	Formicidae	*Oecophylla smaragdina* Fabr.	Satghupu	May–July	all	B, C, CH
	Vespidae	*Parapolybia varia* Fabr.	Pughukhi	May–Sept	all	C, F
		*Polistes olivaceus* De Geer	Pighikhi	May–Sept	all	C, F
		*Polistes stigma* Fabr.	Pughukhi	May–Sept	all	C, F
		**Polistes* sp.	Pughukhi	May–Sept	all	C, F
		*Provespa barthelemyi* du Buysson	Akizu	Sept–Feb	all	C, F
		**Ropalida rufoplagiata* Cameron	Awopu	Sept–Feb	all	C, F
		**Ropalida* sp.	Awopu	Sept–Feb	all	C, F
		*Vespa affinis indosinensis* Perez	Akhighü	Sept–Feb	all	C, F
		*Vespa auraria* Smith	Akhibo	Sept–Feb	all	C, F
		*Vespa basalis* Smith	Akhitsu	May–Nov	all	C, F
		*Vespa bicolor* Fabr.	Akhitsu	May–Nov	all	C, F
		*Vespa ducalis* Smith	Angui	May–Nov	all	C, F
		*Vespa mandarinia* Smith	Akhighü	Sept–Feb	all	C, F
		*Vespa tropica tropica* Linn.	Akichekhi	May–Nov	all	C, F
		*Vespa soror* du Buysson	Akhitsü	Sept–Feb	all	C, F
		***Vespula orbata* du Buysson	Akhibo	May–Nov	all	C, F
		*Vespula* sp.	Akhibo	May–Nov	all	C, F
Lepidoptera	Cossidae	*Cossus* sp.	Aphukulho	July–Feb	larva, pupa	B, C
	Crambidae	*Omphisa fuscidentalis* Hampson	Akhaukhulo	Aug–Oct	larva, pupa	C, F, ST
	Hesperiidae	*Erionata torus* Evans	Wuchoninga	Sept–Nov	larva, pupa	B, C
	Lasiocampidae	*Malacosoma* sp.	Michekhane ninga	Nov–Dec	larva, pupa	C
	Noctuidae	**Pericyma cruegeri* Butler	Katsulanga	April–May	larva	C, F
	Saturniidae	**Antheraea assamensis* Helfer	Muga	Whole year	larva	C
		**Antheraea mylitta* Drury	Bokori ra	May–July	larva	C
		*Samia cynthia ricini* Boisduval	Eri	Whole year	larva, pupa	C, F
Diptera	Tipulidae	*Tipula* sp.	Pochighoh	Sept–Jan	larva	B, C, F

B—boiled; C—cooked; CH—chutney; F—fried; R—raw; RO—roasted; ST—steamed; TO—toasted. * New addition of edible insect in Nagaland. ** New record from India, worker wasp examined, regd. no. ZSI/WGRC/I.R-INV.8972 (P. Girish Kumar and James M. Carpenter, 2018).

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
