# Peer review of "Traditional Knowledge of the Utilization of Edible Insects in Nagaland, North-East India"

_foods, 2020, doi:10.3390/foods9070852_

Round 1
Reviewer 1 Report
Congratulations: I enjoyed reading your excellent report on the uses of insects in Naga culture and cuisine. I have only a few comments.
In the keywords you should italicise the scientific names of the insects, e.g., Samia cynthia ricini, etc.
Line 48: write "....of an earlier investigaton..." (NOT 'our')
Line 50: delete "most"
Line 60: what is the difference between 'animal rearing' and a line further down 'livestock'? Does one refer to poultry and the other to beef or dairy cattle?
Line 66: regarding the 370 informants, can you mention how many of them were male and female? What about the approx. ages of the informants, their main occupation? Were they all literate?
Line 188 and many other places: how exactydo you distinguish or define 'fried' and 'roasted', 'cooked' and 'boiled'?
Line 316: write "...are a natural, renewable resource..."
Line 320: write "However, over-harvesting..." or in one word "...overharvesting..."
Line 327: ...weevil larvae...
Line 334: delete "in" before "keeping"
Reviewer 2 Report
- Materials and Methods
line 62 would you mean that terrace cultivation is fixed as opposed to shifting? Please specify, this aspect is of interest
line 67 Informants are consumers, experts? please clarify
3.Results.
line 106 why insect consumption could be useful for nature? are you referring to the environmental and/or economic sustainability of insect as foods? this aspect is worthy of a deeper explanation
Section 4. Rearing and Marketing of edible insects
As a food for though: how to reconcile the consumption of bee products with their capture for consumption?
- Conclusion and Recommendations
I suggest you to explain how your interesting results can be of interest for a wider audience : community, policy makers, traders….
Reviewer 3 Report
Journal: Foods (ISSN 2304-8158) Manuscript ID: foods-839802 Type: Article Number of Pages: 23 Title: Traditional knowledge of the utilization of edible insects in Nagaland, Northeast India Comments This manuscript describes the existing knowledge of entomophagy in Nagaland, Northeast India. Information on the different insect species (about 106 of them), the method used to process these before consumption, and the degree of domestication of these insects is given. The manuscript is well written. Overall, the content of the manuscript is timely, owing to the renewed interest in entomophagy, ethnic health foods, and diversifying the sources of dietary proteins. The content is also consistent with the aims and scope of this journal. The background information in the Introduction is good. But a description on the role of insects in Nagaland diets is largely missing. In order words, why did authors choose Nagaland? Authors must mention and discuss this briefly in the Introduction. I approve the publication of this manuscript after the following (minor) changes are made. L69 The questionnaire used must be included as Supplementary data. Also, it is not clear whether authors secured ethical clearance for running their questionnaires.The supplementary data in Table S1 is an important piece of information that has to be included in the main text.
Authors mention that there are voucher specimens deposited at the Department of Zoology, Nagaland University. Could authors use those to take high resolution (and perhaps even professional) images, with consistent white background? Although the images given now are good, it is a bit of a challenge to see some distinct features on some of the insects (e.g. Plate 1C; 4B, 6I, 7D,etc.)
L161 Should be "...toasting and eating as..."
L229 and 235. The species names of the bees should be in italics.
The mention of honey in L234 is appropriate. However, because the manuscript is about edible insects and not necessarily about insect products, I think Plates 8D, 8E and 8F can be removed.
L306-308 "Therefore, even though there is potentiality of edible insects to become a staple food, ways for packaging edible insects and increasing their shelf-life to promote them as food nationally and internationally is the need of the hour". This is a good point. But what do authors recommend? Can authors give some ideas of ways in which these insects can be packaged to increase the shelf-life? I assume this is what the people in Nagaland would want to know, as it will increase the chances of commercializing insects.
Could authors make it clear how this manuscript differs from the already published one (i.e. Ref 19, https://www.tandfonline.com/doi/abs/10.1080/09709274.2017.1399632)? And discuss the current manuscript in conjunction with that?
